# Fostering human learning in sequential decision-making: Understanding the role of evaluative feedback

**Piyush Gupta** *, **Subir Biswas, Vaibhav Srivastava**

Department of Electrical and Computer Engineering, Michigan State University, East Lansing, Michigan, United States of America

* guptapi1@msu.edu

**Data Availability Statement:** The data and code is publicly available at https://github.com/piyushgupta221/Decision_making_ToH.

**Funding:** This work has been supported in part by the NSF awards IIS-1734272 and ECCS-2024649.

## Abstract

Cognitive rehabilitation, STEM (science, technology, engineering, and math) skill acquisition, and coaching games such as chess often require tutoring decision-making strategies. The advancement of AI-driven tutoring systems for facilitating human learning requires an understanding of the impact of evaluative feedback on human decision-making and skill development. To this end, we conduct human experiments using Amazon Mechanical Turk to study the influence of evaluative feedback on human decision-making in sequential tasks. In these experiments, participants solve the Tower of Hanoi puzzle and receive AI-generated feedback while solving it. We examine how this feedback affects their learning and skill transfer to related tasks. Additionally, treating humans as noisy optimal agents, we employ maximum entropy inverse reinforcement learning to analyze the effect of feedback on the implicit human reward structure that guides their decision making. Lastly, we explore various computational models to understand how people incorporate evaluative feedback into their decision-making processes. Our findings underscore that humans perceive evaluative feedback as indicative of their long-term strategic success, thus aiding in skill acquisition and transfer in sequential decision-making tasks. Moreover, we demonstrate that evaluative feedback fosters a more structured and organized learning experience compared to learning without feedback. Furthermore, our results indicate that providing intermediate goals alone does not significantly enhance human learning outcomes.

## 1 Introduction

The integration of advanced Artificial Intelligence (AI) algorithms and affordable Internet of Things (IoT) devices has led to the widespread use of these technologies in various personal and professional devices. AI algorithms can handle complex decision-making challenges and support individuals in achieving their learning goals. However, it remains uncertain if embedding intelligent technology in these devices enhances individuals' reasoning and decision-making abilities. To this end, we explore the potential benefits of offering feedback derived from AI's optimal policies in the context of sequential decision-making tasks. Our primary

The analysis portion of this work is supported in part by the ONR award N00014-22-1-2813. The funders had no role in study design, data collection and analysis, decision to publish, or preparation of the manuscript.

**Competing interests:** The authors have declared that no competing interests exist.

goal is to evaluate whether this feedback can effectively enhance an individual's performance in a specific task and whether the acquired knowledge and skills can be readily transferred to related tasks. Through this study, we aim to uncover the influence of AI-generated evaluative feedback on human decision-making.

The examination of the interaction between AI and embodied human intelligence has far-reaching implications for various domains such as cognitive rehabilitation after brain injuries or strokes [1, 2], sports coaching, surgical training, driving instruction and human-supervisory systems [3–5]. The design of automated tutoring systems for assisting humans in learning new tasks has been a topic of significant interest [6–13]. Historically, these systems have been based on the manual coding of domain knowledge, which is then translated into a human-readable format. Recent works [8] have started to explore machine-learning approaches to design automated tutoring systems but do not account for human learning dynamics. Some researchers have also examined the role of cognitive architecture in the design of effective tutoring systems [14, 15], yet these efforts still primarily rely on traditional methods of manual coding of domain knowledge.

In this work, we focus on sequential decision-making tasks [16, 17] that inherently present significant cognitive challenges. They require continual decision-making at each time step, with each choice potentially influencing future states and overall outcomes. These tasks involve navigating the exploration-exploitation trade-off [18–20], which pertains to deciding whether to act based on current knowledge or to explore in order to enhance that knowledge. Proficiency in these tasks can significantly enhance problem-solving skills.

We selected the Tower of Hanoi (ToH) puzzle [21–23] as our choice for the sequential decision-making task. This choice is motivated by the simplicity of the ToH task, enabling efficient learning and evaluation within a reasonable timeframe for our experiment. Nonetheless, it's essential to note that the framework discussed in our work is broadly applicable and can be generalized to other complex sequential decision-making tasks. In ToH, various-sized disks are arranged on three pegs, and the objective is to reach a specific disk configuration by moving one disk at a time. Importantly, only the uppermost disk on a peg can be moved, and larger disks cannot be placed on top of the smaller ones. Decision-making in ToH has been frequently employed in psychological research, serving as a valuable tool for examining developmental progress in children and adolescents [24]. In the cognitive assessment domain, ToH is instrumental for gauging visual-spatial and complex problem-solving capabilities in both adults [25] and children [26]. Solving the ToH task not only requires strong cognitive skills but also relies heavily on executive functions, especially planning [27]. Planning is essential for tackling complex reasoning tasks as it involves controlling impulsive actions and prioritizing strategic problem-solving.

In this study, we explore the impact of AI-generated evaluative feedback on human decision-making, specifically within the context of the ToH puzzle. The AI agent learns the optimal ToH policy and provides evaluative feedback to guide human participants. We evaluate various forms of feedback on decision-making performance and knowledge transfer, conduct experiments to visualize skill development with and without feedback, and investigate models for understanding how humans incorporate feedback into their decision-making processes. This research provides insights into the role of feedback in shaping human decisions.

There are three major contributions to this work.

(i). *Exploring Evaluative Feedback Strategies:* We investigate the impact of different evaluative feedback strategies on the performance of individuals learning to solve ToH, a widely studied sequential decision-making task. Furthermore, we explore how individuals trained with different feedback strategies transfer their skills to a more challenging task.

(ii). *Understanding Reward Structures Induced by Evaluative Feedback:* Treating humans as noisy optimal agents, we study how various evaluative feedback strategies affect their reward functions. Our research highlights the influence of different forms of evaluative feedback on the implicit reward structure that explains human decisions.

(iii). *Developing Computational Models for Human Decision-Making:* We create a set of candidate computational models that may explain how humans integrate evaluative feedback into their sequential decision-making processes. Our goal is to identify the model that best explains human decision-making under evaluative feedback conditions.

The rest of the manuscript is structured as follows. Sec. 2 presents background and problem formulation, and includes a discussion of the ToH structure, the application of maximum entropy IRL for learning human rewards, and the development of computational models aimed at integrating evaluative feedback into human decision-making processes. In Sec. 3, we provide details of the ToH experiments, conducted through the Amazon Mechanical Turk (AMT) platform, alongside the discussion of the various evaluative feedback strategies employed during these experiments. We discuss and analyze the experiment results in Section 4 and finally conclude in Sec 5.

## 2 Background and problem formulation

We investigate the influence of evaluative feedback on human performance in a sequential decision-making task through experimental evaluations and computational modeling. To this end, we conducted experiments, where the participants were asked to solve the ToH puzzle. ToH is a puzzle in which disks with a priority order are placed on three pegs. The priority order determines which disk can be placed on top of another disk and each instance of admissible disk placement is referred to as a configuration. Thus, for a four-disk and a five-disk ToH, there are $3^4 = 81$ and $3^5 = 243$ possible configurations, respectively. The goal is to move one disk at a time and reach the desired configuration while maintaining the priority order at each time.

Consider the ToH puzzle with $n$ disks, where the disks are numbered $\{0, 1, \ldots, n-1\}$ in ascending order of size, and the three pegs are numbered $\{0, 1, 2\}$ from left to right. The state of the $n$-disk ToH can be represented as $S_n = (s_0 s_1 \ldots s_{n-1})$, where $s_i \in \{0, 1, 2\}$ denotes the peg on which disk $i$ is placed, for $0 \leq i \leq n-1$. Each state in an $n$-disk ToH has either two or three possible state transitions as can be seen by the state space of a 4-disk ToH shown in Fig 1.

### 2.1 Evaluative feedback

We train a reinforcement learning (RL) agent [28, 29] that is capable of optimally solving the ToH puzzle. RL is a sub-domain of machine learning aimed at learning an optimal policy in sequential decision-making problems using a reward function. This is achieved by maximizing expected cumulative discounted rewards in each state, also known as the value function. Consider a Markov Decision Process [19]

$$\mathcal{M} = \{\mathcal{S}, \mathcal{A}, \mathcal{T}_{s'}^{sa}, \gamma, \mathbf{r}\}, \tag{1}$$

where $\mathcal{S}$ is the state space, $\mathcal{A}$ is the action space, $\mathcal{T}_{s'}^{sa}$ is the probability of transition from state $s \in \mathcal{S}$ to state $s' \in \mathcal{S}$ under the action $a \in \mathcal{A}$, $\gamma \in [0, 1)$ is the discount factor, and $\mathbf{r} : \mathcal{S} \times \mathcal{A} \to \mathbb{R}$ is the reward function. Let $\mathbf{r}(s_t, a_t)$ be the reward at time $t$ in state $s_t \in \mathcal{S}$ under the action $a_t \in \mathcal{A}$. The agent's actions are defined by its policy $\pi$, where $\pi(a|s)$ is the probability of taking action $a$ in state $s$. The total discounted reward from time step $t$ onwards is referred to as the

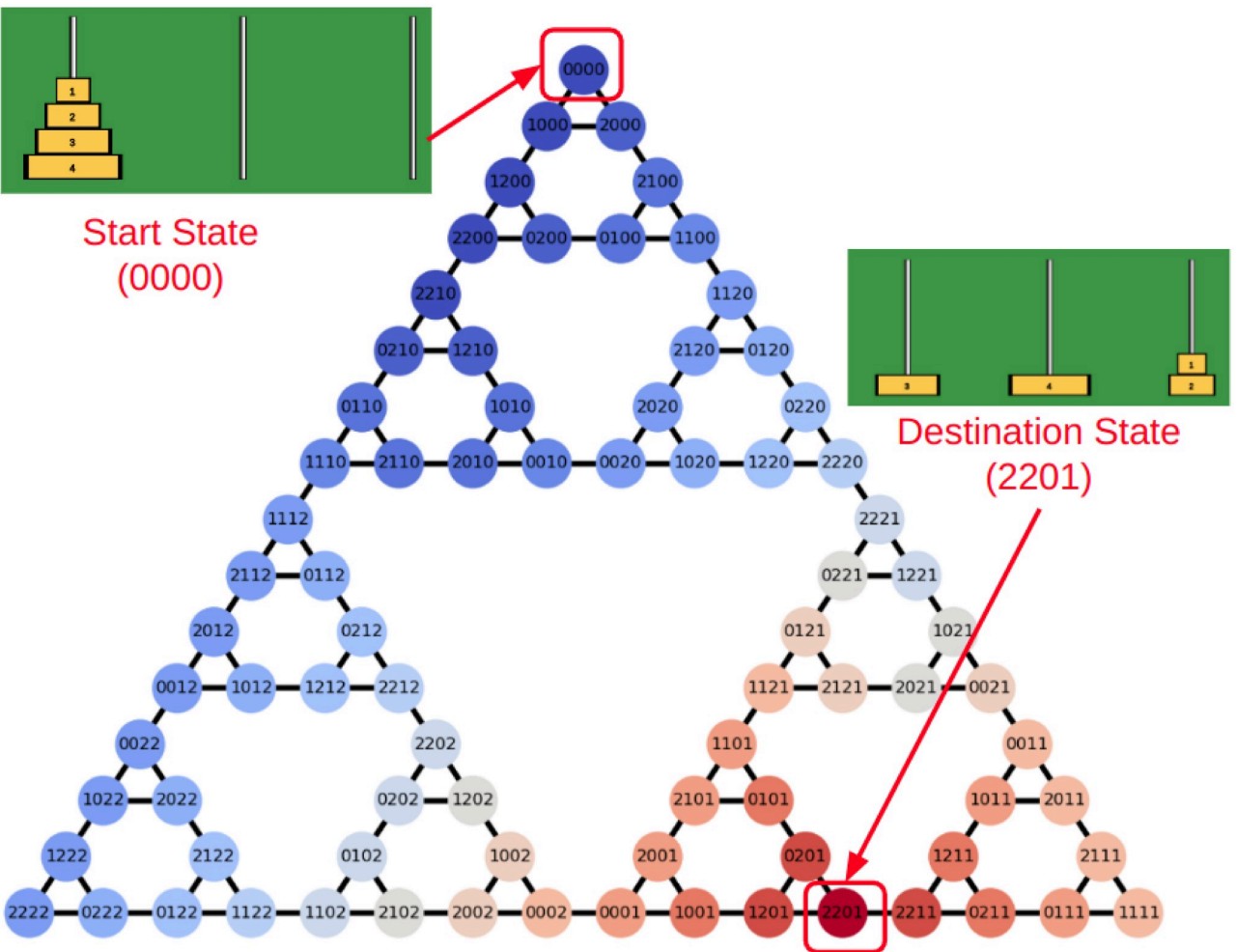

**Fig 1. State space of a 4-disk ToH with 81 states.** Each state corresponds to a unique configuration of the disks on three pegs and edges encode allowed transitions between states. The task is to reach the configuration associated with a randomly selected target state (for example 2201 in this figure). Warmer colors are associated with the higher value function (see Sec. 2.1 for discussion).

return which is defined as:

$$G_t = \sum_{k=t}^{\infty} \gamma^{k-t} \mathbf{r}(s_k, a_k). \tag{2}$$

The expected return of taking action $a$ in the initial state $s$, and subsequently following policy $\pi$, can be quantified by the $Q$ function, which is defined as:

$$Q^{\pi}(s, a) = \mathbb{E}[G_0 | s_0 = s, a_0 = a; \pi], \tag{3}$$

where $s_0$ is the initial state and $a_0$ is the initial action applied. Furthermore, the value of a state (expected return) under a given policy $\pi$ is given by the value function $V$ defined as:

$$V^{\pi}(s) = \mathbb{E}[G_0 | s_0 = s; \pi], \tag{4}$$

where $s_0$ is the initial state. RL algorithms aim to find an optimal policy $\pi^*$ that results in the optimal value function $V^*(s) = \max_a Q^{\pi^*}(s, a) = \max_\pi \mathbb{E}[G_0 | s_0 = s; \pi]$ for each state $s$, i.e., $\pi^*(s) = \operatorname{argmax}_\pi V^\pi(s)$.

The ToH is a finite state space and finite action puzzle, and thus, an optimal policy can be derived using tabular RL methods as described in [28, 30]. In Fig 1, we demonstrate the optimal value function for the standard 4-disk ToH. To obtain the optimal value function for a given target state, we utilize the value iteration algorithm [28], where the reward function $r(s) : \mathcal{S} \to \mathbb{R}$ is designed as follows:

$$r(s) = \begin{cases} 1, & \text{if } s \text{ is the target state}, \\ 0, & \text{otherwise}. \end{cases} \tag{5}$$

Using the reward function in (5) results in an optimal value function that is proportional to the length of the shortest path for each state to the target state. The obtained optimal value function is utilized to provide evaluative feedback to the human player based on the change in the value at states before and after the move. We deploy several feedback mechanisms as detailed in Section 3.1 and systematically explore how human decision-making is influenced by different feedback mechanisms.

**Remark 1** *The value function for each state in the ToH problem is proportional to the shortest path length to the target state, allowing for the application of simpler graph-search algorithms rather than RL. However, it's crucial to recognize that this characteristic is unique to ToH's finite and structured state space, governed by a recursive pattern, and does not apply to all sequential decision-making problems. In complex sequential decision-making problems like chess, characterized by larger or continuous state and action spaces, simpler algorithms might not be available, necessitating the use of advanced AI techniques like RL or deep neural networks to obtain the optimal policy. However, our framework is broadly applicable and can be generalized to other complex sequential decision-making tasks.*

The state space of the ToH problem exhibits a recursive structure. Specifically, the state space of a ToH puzzle with $n$ disks can be effectively illustrated using three interlocking triangles. Each of these triangles symbolizes the state space of a ToH puzzle with $n - 1$ disks. To illustrate this concept, let's examine the state space of a 4-disk ToH in Fig 2, which is highlighted in red. In the same figure, the blue and green squares are employed to represent the state spaces of 3-disk and 2-disk ToH puzzles, respectively. Hence, the state space for the ToH with $n - 1$ disks can be simply achieved by removing the last digit from each state in the upper triangle of the $n$-disk ToH. This digit corresponds to the position of the largest disk.

As illustrated in Fig 2, the state space of the 4-disk ToH puzzle can be decomposed into three triangles labeled as $T_1$, $T_2$, and $T_3$. Throughout the remainder of the manuscript, we will consistently refer to the regions of the state space as follows: the top triangle will be denoted as $T_1$, the lower left triangle as $T_2$, and the lower right triangle as $T_3$. These triangles are interconnected at their vertices through single edges. These vertex states are critical states, transitioning from one triangle to another necessitates passing through these states. For instance, starting from an initial state in $T_1$, the optimal path to reach a desired state in $T_2$ or $T_3$ must involve the state transitions $1110 \to 1112$ and $2220 \to 2221$, respectively. Indeed, to master the art of solving the ToH puzzle effectively, one must grasp its inherent recursive structure. Success in solving the puzzle relies on systematically working towards reaching the crucial critical states within the state space.

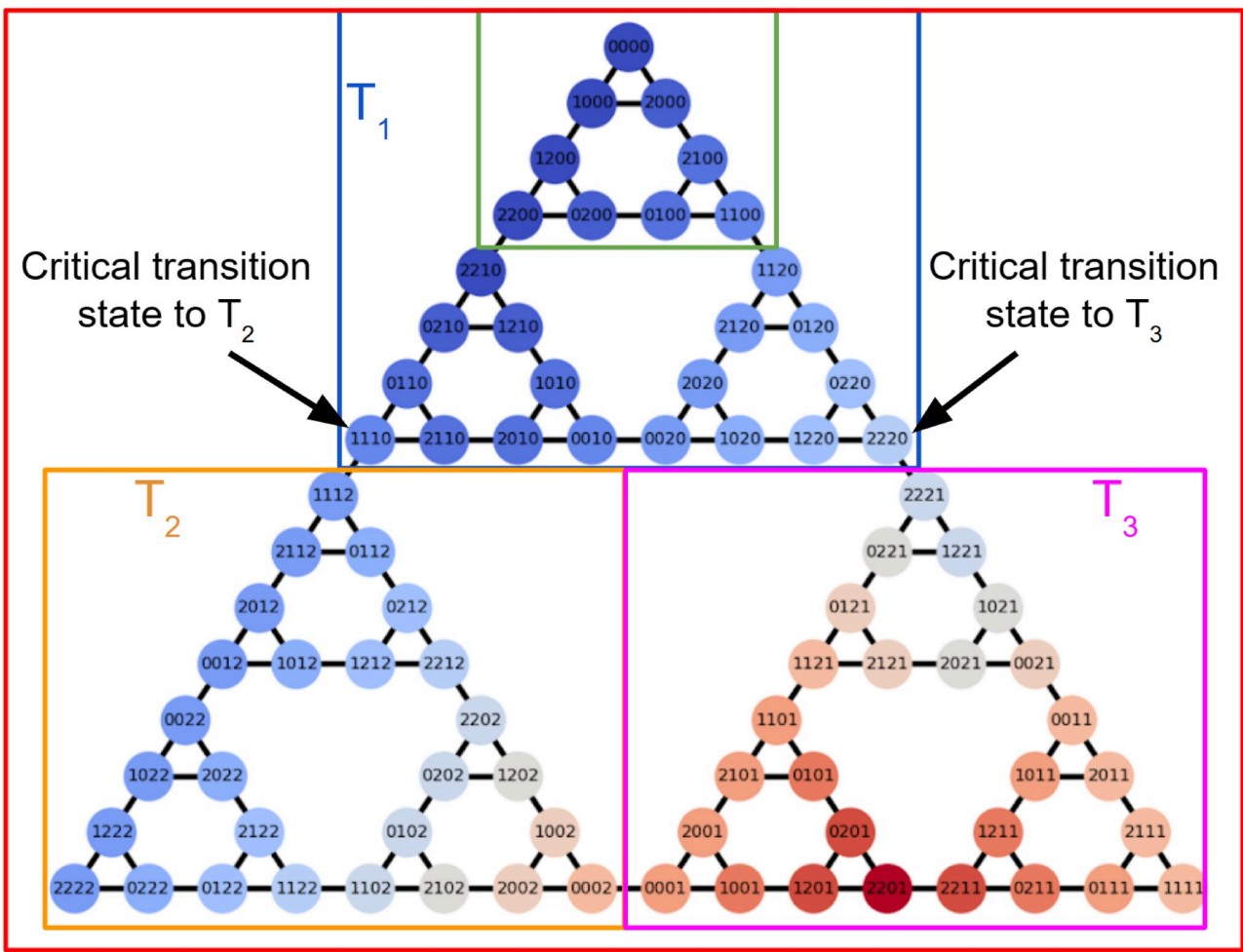

**Fig 2. Recursive structure of the state space of a 4-disk ToH with 81 states.** Each state corresponds to a unique configuration of the disks on three pegs and edges encode allowed transitions between states. The state space can be visualized as comprising three triangular structures. The states that connect different triangular structures are critical states to transition between triangles.

## 2.2 Human rewards using maximum entropy inverse reinforcement learning

In the context of human participants solving the ToH puzzle, we can perceive them as noisy optimal agents striving to optimize an implicit reward function. Utilizing their demonstrations, we can leverage Inverse Reinforcement Learning (IRL) techniques [31, 32] to deduce a reward function. This reward function is designed to align the optimal policy with the observed human demonstrations.

The maximum entropy IRL [33, 34] assumes human demonstrations are not perfect and allows us to learn from sub-optimal demonstrations by incorporating a probabilistic model that captures the variability in human behavior. Maximum entropy IRL has gained significant traction in the literature as a means to effectively learn from human demonstrations [35].

Consider the Markov Decision Process as defined in (1). In the case of the $n$-disk ToH, each edge originating from a given state $s$ in the state transition graph can be considered a unique action. Under these actions, the transition probability $\mathcal{T}_{s'}^{sa}$ equals 1 for the transition

from state $s$ to $s'$ if they are connected through an edge. Let $\mathcal{D} = \{\zeta_1, \ldots, \zeta_N\}$ be the set of $N$ demonstrations, where each demonstration $\zeta_i$ is a path $\zeta_i = \{(s_{i,0}, a_{i,0}), \ldots, (s_{i,T}, a_{i,T})\}$. The unknown reward function $\mathbf{r}$ is expressed as a linear combination of a set of predefined features denoted as $f : \mathcal{S} \to \mathbb{R}$. The weights associated with these features are learned through the maximum entropy IRL algorithm.

In the framework of maximum entropy IRL, the probability of following a particular trajectory $\zeta$ is directly proportional to the exponential of the accumulated rewards experienced along that path. This leads to a stochastic behavior model, where the probability of taking a specific action $a$ in a given state $s$ is determined by the exponential of the expected total reward subsequent to taking that action, i.e., $P(a|s) \propto \exp(Q_{sa}^r)$, where $Q_{sa}^r$ is computed as $Q_{sa}^r = \mathbf{r} + \gamma \mathcal{T} V_s^r$. The value function $V_s^r$ is computed using a "soft" variant of the familiar Bellman operator [28]: $V_s^r = \log \sum_a \exp(Q_{sa}^r)$. Consequently, the probability of action $a$ in state $s$ is normalized by $\exp(V_s^r)$, yielding $P(a|s) = \exp(Q_{sa}^r - V_s^r)$.

The complete log-likelihood of the observed data under the reward function $r$ can be expressed as:

$$\log P(\mathcal{D}|\mathbf{r}) = \sum_i \sum_t \log P(a_{i,t}|s_{i,t}) = \sum_i \sum_t (Q_{s_{i,t} a_{i,t}}^r - V_{s_{i,t}}^r). \tag{6}$$

Interested readers are referred to [35] for detailed derivations.

We employ maximum entropy IRL to infer the reward functions associated with human behavior. Detailed results are presented in Section 4.2.

## 2.3 Modeling human sequential decision-making under feedback

A central and challenging aspect of designing efficient tutoring systems lies in understanding the impact of evaluative feedback from AI on human decision-making. Precisely, the question of how humans incorporate feedback into their decision-making processes is of paramount importance. In modeling this process, a foundational challenge is understanding how they interpret the feedback, including whether it's seen as an immediate reward or an evaluation of long-term impacts. Further, it's important to explore if feedback relates only to the current action or spans the sequence of actions. Additionally, understanding if evaluative feedback affects the assessment of value functions over time or momentarily influences action choices is vital. To tackle these questions, we develop candidate models that embody different mechanisms for incorporating feedback into human decision-making processes. Our models are inspired by the Training an Agent Manually via Evaluative Reinforcement (TAMER) framework [36–39] developed to incorporate human feedback into the policy of an artificial RL agent.

Let $\hat{H}(s, a)$ and $|f|$ denote the evaluative feedback and number of predefined features $f$ employed to define human rewards, respectively. We study four different models detailed as follows:

- Model 1—Ignore feedback: This baseline model operates under the assumption that evaluative feedback isn't directly integrated into human decision-making processes. Instead, individuals are postulated to focus on maximizing the long-term value derived from their personal reward functions. In this framework, evaluative feedback plays an indirect role by shaping and refining these reward functions. This is the default model studied in Sec. 2.2. The model encompasses $|f|$ learned parameters.

- Model 2—Update $Q(s, a)$: In this model, we postulate that humans interpret evaluative feedback as an indicator of the long-term effectiveness of their strategic actions, serving as an

approximation of $Q_{sa}^r$. The model integrates this feedback to update the Q-estimate as follows:

$$Q'(s, a) = Q(s, a) + k\hat{H}(s, a), \tag{7}$$

where $k$ is a parameter to be learned. Consequently, the policy gets updated as $P(a|s) = \exp(Q_{sa}' - V_s')$, where $V_s' = \log \sum_a \exp(Q_{sa}')$ denotes the newly adjusted value function. The model encompasses $|f| + 1$ learned parameters.

- Model 3—Update $r(s, a)$: In this model, we postulate that humans perceive the evaluative feedback as a measure of the myopic effectiveness of the strategy, serving as an approximation of $r(s, a)$. The model updates the human rewards as follows:

$$r'(s, a) = r(s, a) + k\hat{H}(s, a), \tag{8}$$

where $k$ is a parameter to be learned. The updated reward function is used to estimate the Q-values and the policy. The model encompasses $|f| + 1$ learned parameters.

- Model 4—Feedback as a measure of $Q(s, a)$: In this model, we assume that humans ignore learning by interaction and treat evaluative feedback as a fixed measure of $Q(s, a)$. Therefore,

$$Q'(s, a) = k\hat{H}(s, a), \tag{9}$$

where $k$ is the only parameter to be learned.

**Remark 2** *It's worth noting that the log-likelihood of the maximum entropy IRL depends solely on Q(s, a) through P(a|s). Consequently, Model 2 can be considered equivalent to another model where humans utilize feedback solely to influence their action selection. In this alternate model, humans do not incorporate evaluative feedback into their estimation of Q(s, a); rather, they use it exclusively to bias their action selection, i.e., P(a|s) ∝ exp(Q(s, a) + kĤ(s, a)). In the context of maximum entropy IRL, this model is tantamount to Model 2, where humans employ evaluative feedback to update their Q-estimates as Q'(s, a) = Q(s, a) + kĤ(s, a).*

We investigate these models in Sec 4.3 to understand how humans incorporate evaluative feedback in their decision-making.

## 3 Human experiments

In this section, we discuss the human experiments conducted using AMT.

### 3.1 Experiment design

We examine the effect of evaluative feedback on sequential decision-making using ToH task. To achieve this, we designed five separate experiments, each featuring a different type of feedback. Participants for each experiment were recruited randomly through AMT. Each participant first solved a 4-disk ToH task ten times (training task) and then a 5-disk ToH task five times (transfer task) to evaluate their skill transfer to a more challenging task. The initial state of each puzzle was standardized with all the disks located on the first peg. Considering the state-space of the puzzle as comprising of three interlocking triangles, the target state was randomly selected from the states within the triangles that did not include the initial state, i.e., triangles $T_2$ and $T_3$. The participants were given a maximum number of moves $m_{\text{allowed}}$ to solve the puzzle, calculated as:

$$m_{\text{allowed}} = \lceil 1.5 \times m_{\text{min}} \rceil, \tag{10}$$

where $m_{\text{min}}$ represents the minimum number of moves from the initial configuration to the

final configuration, as determined by the minimum path length in the state graph (Fig 1). While we do not impose specific time limits for individual tasks, there was an overall time limit of 90 minutes allocated for completing all the training and transfer tasks. It's worth mentioning that all participants successfully completed their tasks within this stipulated time frame.

The only difference among the experiments was the feedback provided during the 4-disk ToH training task. No feedback was provided during the 5-disk ToH transfer task in any of the experiments. In each experiment, participants were asked to try their best to get the highest scores. The feedback and scoring metrics used during the training task for the five experiments were:

(i). *Experiment 1—No feedback:* In this experiment, the participants solved the 4-disk ToH puzzle without any feedback. The scoring metric for these tasks was selected as:

$$S = 10(m_{\text{allowed}} - m_{\text{used}} + 1), \tag{11}$$

where $m_{\text{used}}$ is the total moves used to solve the puzzle. The participant receives a score of 0 if the puzzle remains unsolved after exhausting the allowed number of moves. The same scoring metric was used in the 5-disk transfer tasks in all the experiments.

(ii). *Experiment 2—Numeric feedback:* The participants in this experiment received visual feedback on each move they made while solving the 4-disk ToH. The feedback was in the form of text that reads "good move + 2" or "bad move −2", indicating whether the move increased or decreased the value of the state (recall that the value of a state is proportional to the minimum path length of that state to the target state), respectively. The scoring metric for the training tasks in this experiment was selected as:

$$S = 10(m_{\text{allowed}} - m_{\text{used}} + 1) + 2(m_{\text{good}} - m_{\text{bad}}), \tag{12}$$

where $m_{\text{good}} \in \mathbb{N}$ and $m_{\text{bad}} \in \mathbb{N}$ denote the number of good and bad moves, respectively.

(iii). *Experiment 3—Optional feedback:* In this experiment, the participants did not receive visual feedback automatically but had the option to request it by pressing a button, which came at the cost of a small penalty. If the participant requested feedback, they would receive the same visual feedback as in Experiment 2, which evaluated their last move. The scoring metric for the training tasks in this experiment was as follows:

$$S = 10(m_{\text{allowed}} - m_{\text{used}} + 1) - f_{\text{optional}}, \tag{13}$$

where $f_{\text{optional}} \in \mathbb{N}$ denotes the number of times the participant requests feedback.

(iv). *Experiment 4—Sub-goal:* In the state graph of the 4-disk ToH task (as illustrated in Fig 1), states 1110 and 2220 are critical in reaching the target states efficiently in triangles $T_2$ and $T_3$, respectively. In this experiment, based on the target configuration, the participants were presented with an intermediate sub-goal configuration (1110 or 2220) in addition to the target configuration. The participant was instructed to try to reach the intermediate sub-goal first. The scoring metric for the training tasks in this experiment was as follows:

$$S = 10(m_{\text{allowed}} - m_{\text{used}} + 1) + 5s_{\text{subgoal}}, \tag{14}$$

where $s_{\text{subgoal}} \in \{0, 1\}$ was set to 1 if the participant successfully reaches intermediate sub-goal configuration, and 0 otherwise.

(v). *Experiment 5—Sub-goal with numeric feedback:* In this experiment, the participants received both the visual feedback as in Experiment 2 and the intermediate sub-goal configuration as in Experiment 4. The scoring metric for the training tasks in this experiment was calculated as follows:

$$S = 10(m_{\text{allowed}} - m_{\text{used}} + 1) + 2(m_{\text{good}} - m_{\text{bad}}) + 5s_{\text{subgoal}}. \qquad (15)$$

This experiment provided the participants with the maximum amount of evaluative feedback.

Fig 3 shows the experimental interface utilized by participants during the training task of Experiment 5. As illustrated in Fig 3, the participants had access to the numeric feedback, the number of moves taken, the current score $S$ (total reward), the maximum available moves $m_{\text{allowed}}$, the maximum possible reward, as well as information regarding intermediate and final goal configurations.

The interface for the training tasks in the other experiments is similar to Experiment 5 with the following key differences with respect to Fig 3:

(i). Experiment 1: Participants do not receive any numeric feedback, like "Bad Move: -2 (in Fig 3)", and intermediate goal configurations during the training tasks.

(ii). Experiment 2: Participants receive numeric feedback during training tasks, but no intermediate goal configurations are provided.

(iii). Experiment 3: Participants were given access to a button labeled "Get Feedback" during the training tasks. Numeric feedback is only provided upon user request by pressing the designated button.

**Tower of Hanoi**

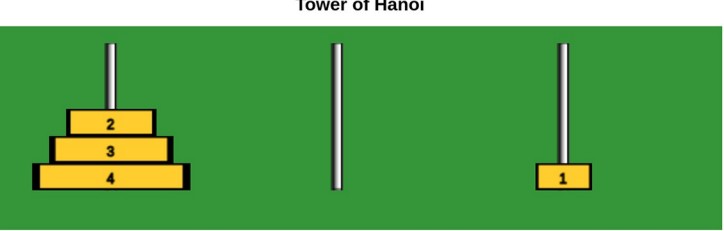

## Bad Move: -2

**Moves: 1   Total Reward: -2**

**Maximum Available Moves: 17     Maximum Possible Reward: 95**

**Intermediate Goal Configuration**

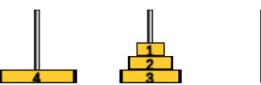

**Final Target Configuration**



**Fig 3. Experimental interface for the human subject participating in the training task of Experiment 5.**

(iv). Experiment 4: Participants receive intermediate goal configurations during training tasks, but no numeric feedback is provided.

## 3.2 Methods

After receiving the IRB consent (MSU IRB #8421) from Michigan State University's IRB office, we recruited 238 participants using AMT for the study. Inclusion criteria were established as having completed a minimum of 500 prior studies and maintaining a 98% approval rate on the platform. Participants were compensated with a base payment of $6 and had the opportunity to earn additional performance-based bonuses ranging from $0 − $4. Of the recruited participants, 78 participants were excluded due to self-reported prior experience with the ToH task.

The recruitment of participants took place from July 3, 2023, to July 10, 2023. Before engaging in the experiment, each participant was required to give written informed consent online, which was then securely documented alongside their experimental data. Participation was restricted to individuals who were 18 years of age or older.

# 4 Results and discussion

In this section, we discuss the results of the experiments conducted on AMT.

## 4.1 Performance under evaluative feedback

First, we collect the data of 20 participants each for the 5 set of experiments detailed in Section 3.1.

In every experiment, we assess participants' performance by calculating their percentage scores for both the training and transfer tasks as follows:

$$100 \times \left( \frac{m_{\text{allowed}} - m_{\text{used}} + 1}{m_{\text{allowed}} - m_{\text{min}} + 1} \right). \tag{16}$$

The maximum percentage score of 100% is achieved when $m_{\text{used}} = m_{\text{min}}$. The minimum 0% is achieved when the puzzle is still unsolved after $m_{\text{used}} = m_{\text{allowed}}$ and $m_{\text{used}}$ transitions to $m_{\text{allowed}} + 1$. The minimum non-zero percentage score of $100 \times \frac{1}{m_{\text{allowed}} - m_{\text{min}} + 1}$ is received by the participant when a puzzle is solved exactly after $m_{\text{used}} = m_{\text{allowed}}$ number of moves. Since $m_{\text{allowed}}$, $m_{\text{min}}$, and $m_{\text{used}}$ are discrete parameters but depend on the experimental setup, the percentage score can take a large number of discrete values between 0% and 100%. Therefore, we employ box plots for visualization for effectively illustrating key quantiles and extrema in the data distribution.

In Fig 4a, we present box plots illustrating the percentage scores achieved in the training tasks (4-disk ToH). Notably, participants who underwent training with evaluative feedback in Experiment 2 (numeric feedback) and Experiment 5 (sub-goal with numeric feedback) exhibited significantly improved performance during these training tasks compared to participants in Experiment 1 (no feedback), who received no evaluative feedback.

In Experiment 3 (optional feedback), participants seldom requested feedback to avoid the feedback penalty, resulting in performance levels akin to those observed in Experiment 1. Experiment 4 (sub-goal) introduced a unique approach, where participants were exclusively exposed to sub-goal configuration (1110 or 2220) crucial for reaching the desired target state. In the absence of evaluative feedback, this method resembled the conditions of Experiment 1, where the sub-goal can be effectively thought as a target state until the sub-goal state is reached. We hypothesize that supplying solely sub-goal configurations without evaluative feedback may

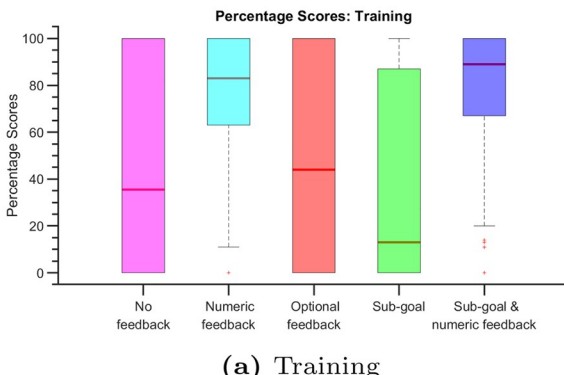
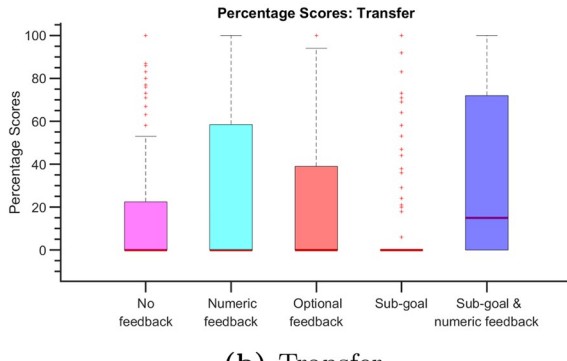

**(a)** Training  **(b)** Transfer

**Fig 4. Box plots displaying percentage scores for both training (a) and transfer (b) tasks.** Within each box plot, the median is represented by the red horizontal line, while the lower and upper edges of the box signify the 25th and 75th percentiles, respectively. Whiskers extend to encompass the most extreme data points that are not classified as outliers, and individual outliers are plotted using the symbol '+'.

induce confusion, as participants may now consider two target states simultaneously—the sub-goal and the target state. Consequently, participants in Experiment 4 exhibited a marginal decrease in performance compared to those in Experiment 1.

In Fig 4b, we present box plots illustrating the percentage scores achieved in the transfer tasks involving the 5-disk ToH. It's important to note that solving the 5-disk ToH, with its 243 states, presents a significantly greater challenge compared to the training task, which involved the 4-disk ToH with 81 states. Furthermore, participants had no prior experience with the 5-disk ToH and relied solely on their training with the 4-disk ToH. Consequently, the transfer tasks yielded relatively lower scores, with many trials failing to solve the puzzle within the allotted number of moves, which can make it challenging to interpret the box plots in Fig 4b.

To focus on successful outcomes, we filtered for positive percentage scores in each experiment, representing the trials where participants successfully solved the ToH puzzle. Table 1 provides an overview of the percentage of successful trials for each experiment, both in the training and transfer tasks. Notably, Experiment 2 and Experiment 5 demonstrated a substantial improvement in successful trials, showing increases of 33.5% and 36%, respectively, compared to Experiment 1 in the training tasks. In the transfer tasks, Experiments 2 and 5 also showed notable improvements, with success rates increasing by 13% and 26%, respectively, compared to Experiment 1.

To assess the statistical significance of these findings, we conducted a two-sample $t$-test comparing the results of Experiments 2 and 5 with the data from Experiment 1. Remarkably, the $p$ values for Experiment 2 (in comparison to Experiment 1) and Experiment 5 (relative to Experiment 1) are $1.59 \times 10^{-12}$ and $1.71 \times 10^{-17}$, respectively, in the training tasks, indicating highly significant differences. In the transfer tasks, the $p$ values are $3.9 \times 10^{-2}$ and $7.17 \times 10^{-4}$ for Experiments 2 and 5 compared to Experiment 1, respectively. Consistent with the commonly accepted significance level of 0.05, a $p$ value below this threshold leads us to reject the null hypothesis, indicating that the data from the two experiments do not arise from the same

**Table 1. Percentage of successful trials in the training and transfer tasks.**

|  | No feedback | Numeric feedback | Optional feedback | Sub-goal | Sub-goal with numeric feedback |
|---|---|---|---|---|---|
| **Training** | 58% | 91.5% | 62% | 52.5% | 94% |
| **Transfer** | 28% | 41% | 35% | 22% | 54% |

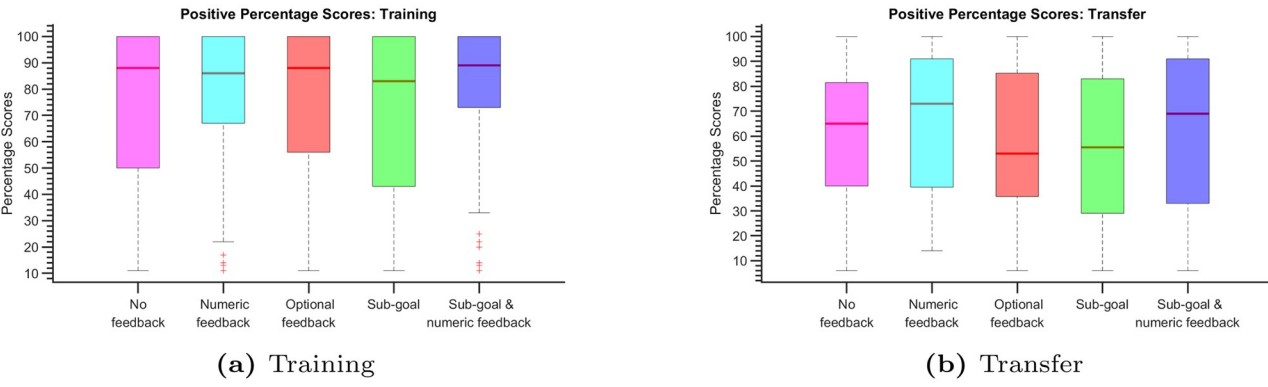

**Fig 5. Box plots displaying positive percentage scores for both training (a) and transfer (b) tasks.**

distribution at a 5% significance level. These results underscore the substantial impact of evaluative feedback on performance, both in the training and transfer tasks.

Fig 5a and 5b display box plots representing successful trials after filtering for positive scores. Notably, the medians of these box plots closely align with each other, suggesting that participants' performances in the experiments can be effectively compared solely through the percentage of successful trials. Once participants have successfully learned to solve the ToH puzzle, their scores exhibit relatively little variation across experiments during successful trials. This observation highlights the stability and consistency of participants' performance once they have mastered the task.

Recall that each participant completed 10 trials of training and 5 trials of transfer tasks. In Fig 6a and 6b, bar plots represent the mean percentage scores for different trials in the training and transfer tasks, respectively. It's evident that participants who received no feedback exhibited relatively low scores compared to those who received either numeric feedback or sub-goals with numeric feedback. Furthermore, while there is no consistent improvement over the trials for participants who did not receive feedback, participants who received evaluative feedback demonstrated performance enhancement with increasing scores across trials. Similar trends are observable in the transfer tasks, indicating that participants who received evaluative feedback found it easier to transfer their skills to related tasks and showed improvement across trials.

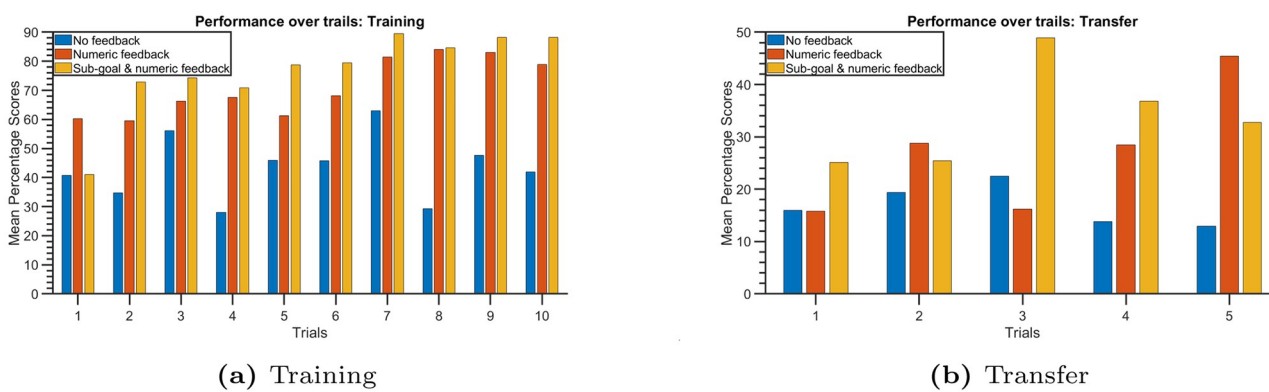

**Fig 6. Bar plots displaying the mean percentage scores for different trials for both training (a) and transfer (b) tasks.**

The results in Table 1 underscore significant improvements in human decision-making attributed to evaluative feedback during training tasks, along with effective skill transfer to related tasks. We employ maximum entropy IRL [34] to investigate the pivotal role of evaluative feedback in shaping human decision-making, as detailed in Sections 4.2 and 4.3. To enable this analysis, we conducted additional data collection sessions with 20 participants each, encompassing experiments devoid of feedback (Experiment 1) and those involving evaluative feedback (Experiments 2 and 5).

## 4.2 Human rewards under evaluative feedback

In this section, we treat humans solving the ToH puzzle as noisy optimal agents striving for optimal play with some implicit reward structure. We examine participants from three sets of experiments: (a) No feedback (Experiment 1), (b) Numeric feedback (Experiment 2), and (c) Sub-goal with numeric feedback (Experiment 5). To gain insights into human learning under these varying feedback conditions, we employ maximum entropy IRL analysis to uncover the underlying human reward structures. Visualizing these human rewards can offer valuable insights into the learning process with and without evaluative feedback.

Recall that for each experiment, the initial state is standardized with the starting state represented by the top vertex of triangle $T_1$, and the target state is randomly selected from either triangles $T_2$ or $T_3$ (see Fig 2). For each experiment, we partition the experimental data into two sets, one with target states in $T_2$ and the other with target states in $T_3$. In each of these sets, we learn the human rewards expressed as a linear combination of predefined features. By modifying these predefined features, we consider two different settings where the human rewards are learned for all the states and for a subset of 8 states. To estimate the rewards, we maximize the log-likelihood as defined in (6), while applying an $\mathcal{L}_1$ penalty to promote sparse rewards. To determine the coefficient of the $\mathcal{L}_1$ penalty $\lambda \in \mathbb{R}_{\geq 0}$, we consider $\lambda \in \{0, 0.1, \ldots, 2\}$ and perform 5-fold cross-validation on the data and select the coefficient that yields the maximum mean log-likelihood across the 5-fold validation sets.

Fig 7a and 7b display the learned IRL rewards in the training tasks for all states. While IRL typically assumes expert demonstrations, it's important to note that participants may still be learning the task during the initial trials. Since the performance does not vary significantly in the latter half of the trials (see Fig 6a), we assume that the human rewards are relatively stationary from trials 6 to 10 and, therefore, exclusively utilize these trials for our IRL analysis. From these latter trajectories, we derive IRL rewards, considering both (a) all available trajectories and (b) only the successful ones, where success is defined by reaching the target state.

Each of these plots is organized into a grid with 2 rows and 3 columns. The top row represents trajectories with the target state in triangle $T_2$, while the bottom row represents trajectories with the target state in triangle $T_3$. The columns correspond to the three sets of experiments: no feedback, numerical feedback, and sub-goal with numerical feedback arranged from left to right.

In Fig 7a, it becomes apparent that participants' rewards in the experiment with no feedback (first column) exhibit a distribution across all states, encompassing both $T_2$ and $T_3$, despite the target state's placement in $T_2$ for the first row and in $T_3$ for the second row. The occurrence of high rewards in $T_3$ (respectively $T_2$) when the target state resides in $T_2$ (respectively $T_3$) primarily stems from the unsuccessful attempts to solve the ToH puzzle in each experiment. Consequently, we observe that as participants' performance improves across experiments from left to right, rewards increase within the triangle containing the target state while decreasing in the opposing triangle. Another noteworthy observation is the presence of high rewards at the critical states (vertices of the target triangle), which serve as pivotal entry

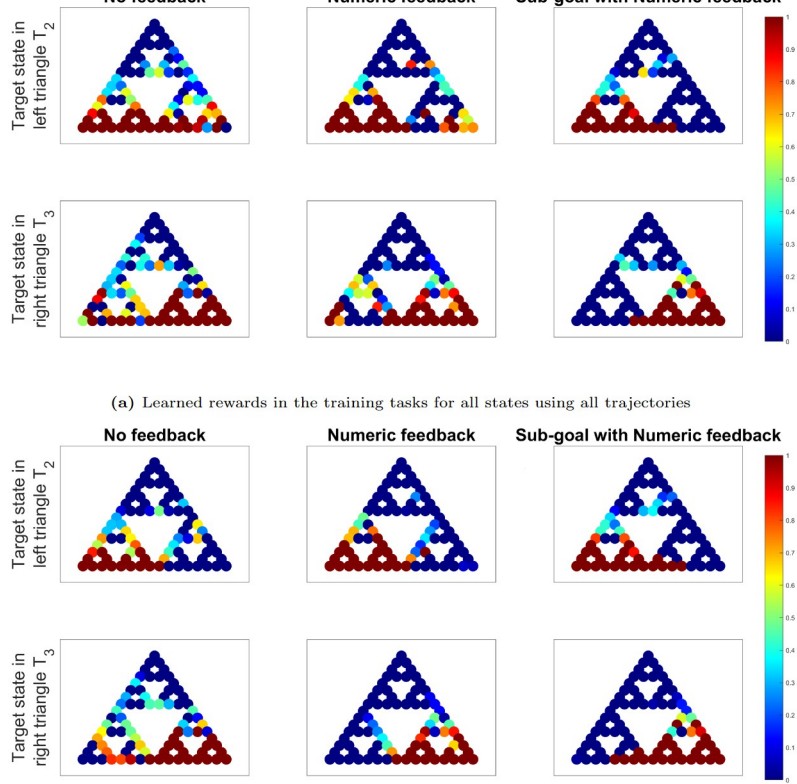

(a) Learned rewards in the training tasks for all states using all trajectories

(b) Learned rewards in the training tasks for all states using only successful trajectories

**Fig 7. IRL plots in training tasks for all states.** IRL plots displaying learned human rewards in the training tasks for all states, using trajectory datasets (from trials 6-10 for each participant) from each experiment that encompass (a) all available trajectories and (b) only successful trajectories, where success is defined by reaching the target state. The red color represents high rewards close to 1 and dark blue represents close to 0 reward.

points to the target triangle. These rewards become more pronounced as performance enhances from left to right.

Fig 7b depicts the learned IRL rewards derived exclusively from successful trajectories in each experiment. Due to the absence of failed trajectories in each experiment, the disparities in IRL rewards across experiments, from left to right, become less pronounced. In each experiment, states within the target triangle and critical states exhibit higher rewards compared to the opposing triangles. In Experiments 1 and 2, the elevated rewards along the edge in the opposite triangle, which is closer to the target triangle, suggest that participants in these experiments occasionally complete the puzzle by opting for suboptimal routes. In contrast, participants in Experiment 5 predominantly solve the puzzle utilizing the optimal trajectory.

Fig 8a and 8b present the learned IRL rewards for all states within the transfer tasks, utilizing trajectory datasets that encompass (a) all available trajectories and (b) only successful trajectories. It is important to note that the transfer tasks pose significant challenges, with none of the participants receiving any feedback. Consequently, the trajectories for the transfer tasks in each experiment comprise numerous failed trajectories.

However, a noticeable trend emerges: participants from Experiment 5, who were trained using sub-goals with numeric feedback, exhibit faster learning in solving the transfer tasks

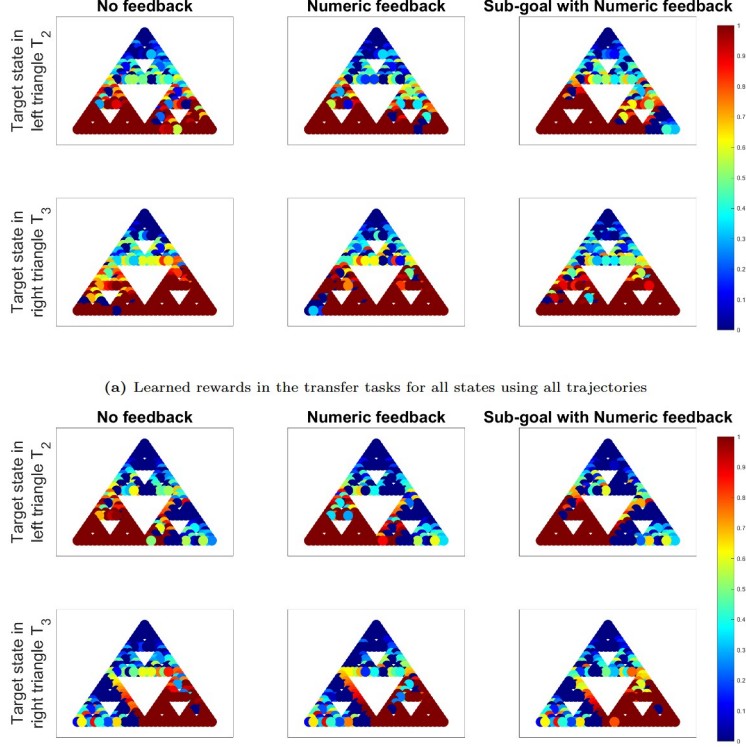

**(a)** Learned rewards in the transfer tasks for all states using all trajectories

**(b)** Learned rewards in the transfer tasks for all states using only successful trajectories

**Fig 8. IRL plots in transfer tasks for all states.** IRL plots displaying learned human rewards in the transfer tasks for all states, using trajectory datasets from each experiment that encompass (a) all available trajectories and (b) only successful trajectories, where success is defined by reaching the target state. The red color represents high rewards close to 1 and dark blue represents close to 0 reward.

compared to participants from Experiments 1 and 2, who received no feedback and only numerical feedback, respectively. This is evident from the higher rewards within the target triangle and lower rewards in the opposite triangle for Experiment 5. When considering only successful trajectories to derive the IRL rewards in Fig 8b, the differences across experiments become less pronounced due to the exclusion of failed trajectories in all experiments.

The results presented in Figs 7 and 8 offer valuable insights into how humans acquire puzzle-solving skills under various evaluative feedback strategies. However, it's worth noting that the learned rewards appear less sparse due to the predefined features, which permit non-zero rewards in all states. Consequently, while these learned IRL rewards for all states offer insights into critical states, they can complicate the comparison between experiments. Furthermore, most of the RL rewards are often sparse. To this end, we modify the predefined features to encourage sparser rewards, allowing non-zero rewards in only 8 states for both the training and transfer tasks. These 8 states were thoughtfully selected as the vertices of the smaller triangles within the state space. In Fig 2, these states correspond to 2200, 1100, 1110, 2220, 0012, 2212, 1121, 0021.

Fig 9a and 9b illustrate the learned IRL rewards for a specific subset of 8 states during the training tasks. These rewards are derived from trajectory datasets obtained from the latter half of the trials (trials 6 to 10) for each participant. We consider two scenarios: (a) using all available trajectories and (b) using only the trajectories that resulted in successful task completion.

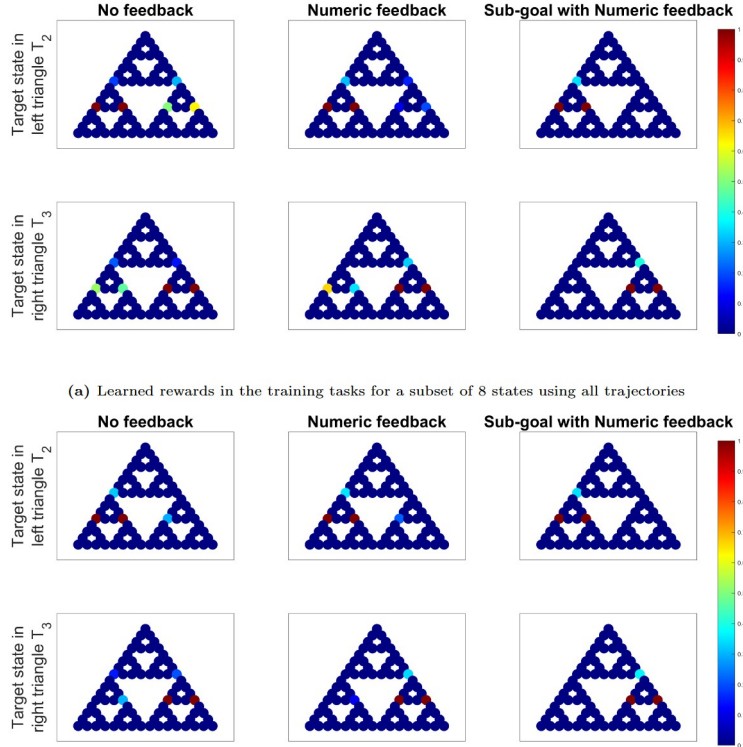

(a) Learned rewards in the training tasks for a subset of 8 states using all trajectories

(b) Learned rewards in the training tasks for a subset of 8 states using only successful trajectories

**Fig 9. IRL plots in training tasks for a subset of 8 states.** IRL plots displaying learned human rewards in the training tasks for a subset of 8 states, using trajectory datasets (from trials 6-10 for each participant) from each experiment that encompass (a) all available trajectories and (b) only successful trajectories, where success is defined by reaching the target state. The red color represents high rewards close to 1 and dark blue represents close to 0 reward.

It is evident that participants from Experiment 5 demonstrate non-zero rewards exclusively within the target triangle and the corresponding critical states. As we progress from left to right, the non-zero rewards in the opposite triangle diminish due to fewer instances of failure. These differences become less pronounced when we solely consider successful trajectories in Fig 9b.

Fig 10a and 10b depict the learned IRL rewards for a selected subset of 8 states within the transfer tasks, using trajectory datasets that encompass (a) all available trajectories and (b) only successful trajectories. While the distinctions are somewhat less pronounced due to the presence of numerous failure attempts in all experiments, the lower rewards in the opposite triangle indicate swifter learning when participants are trained with feedback, in contrast to participants who receive no feedback. These differences become less noticeable when we exclusively consider successful trajectories in Fig 10b, effectively eliminating most of the non-zero rewards in the opposite triangle.

The results of the max entropy IRL analysis underscore the significance of critical states and demonstrate how human learning in sequential decision-making tasks can be organized more effectively when evaluative feedback is provided, in contrast to participants solely learning through exploration without any feedback. The results further indicate that the participants trained with evaluative feedback exhibit an ability to transfer their learning to newer, related, and more demanding tasks at a significantly accelerated pace compared to those who learn without feedback.

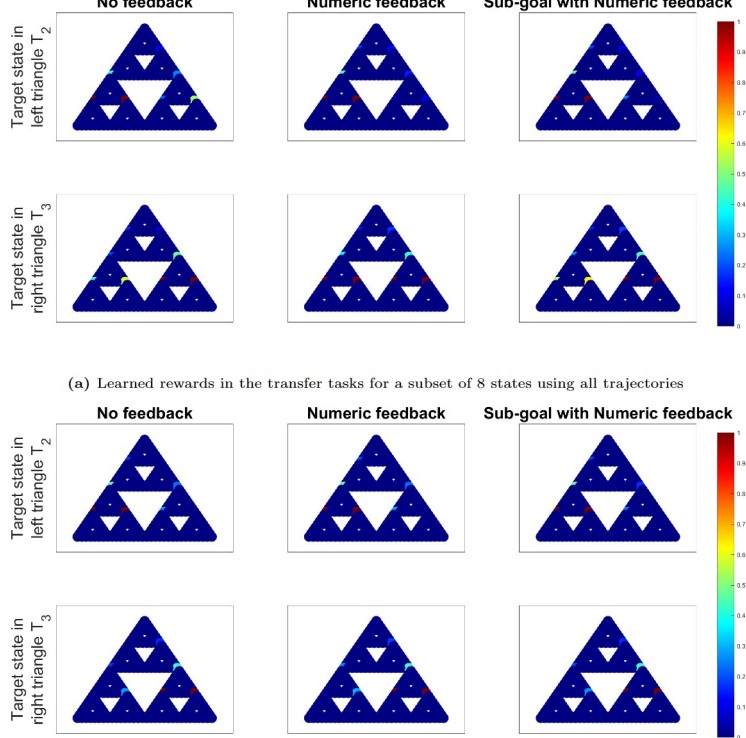

(a) Learned rewards in the transfer tasks for a subset of 8 states using all trajectories

(b) Learned rewards in the transfer tasks for a subset of 8 states using only successful trajectories

**Fig 10. IRL plots in transfer tasks for a subset of 8 states.** IRL plots displaying learned human rewards in the transfer tasks for a subset of 8 states, using trajectory datasets from each experiment that encompass (a) all available trajectories and (b) only successful trajectories, where success is defined by reaching the target state. The red color represents high rewards close to 1 and dark blue represents close to 0 reward.

## 4.3 Modeling human decision-making under evaluative feedback

In Sec. 4.1 and 4.2, we have demonstrated the pivotal role of evaluative feedback in enhancing learning and performance within the context of the ToH puzzle. In this section, we delve into exploring models that aim to elucidate the mechanisms through which humans integrate evaluative feedback into their decision-making processes.

In our analysis, we explore four distinct models for incorporating feedback into human decision-making, as detailed in Sec. 2.3. For each of these models, we calculate both the Akaike information criterion (AIC) [40] and Bayesian information criterion (BIC) [41] to identify the most suitable model. For a given model, the AIC and BIC are defined as:

$$\text{AIC} = 2p - 2\log\left(\hat{L}\right), \quad \text{BIC} = p\log\left(o\right) - 2\log\left(\hat{L}\right), \tag{17}$$

where $p$, $o$, and $\hat{L}$ denote the number of learned parameters, the number of observations, i.e., the sample size, and the maximized value of the likelihood function of the model, respectively. The model with the lowest AIC (or BIC) is deemed the optimal choice according to AIC (or BIC) criteria. For this analysis, we leverage the experimental data gathered during the training tasks of Experiment 2, where participants received numeric feedback.

It is important to note that this numeric feedback is determined based on the change in state value before and after the state transition. Consequently, it is intrinsically tied to the target state, given that the value function is contingent upon the target state.

Since the target state is subject to randomization in triangles $T_2$ and $T_3$, we further segment these triangles into three sub-triangles each. This subdivision allows us to categorize the experimental data into six distinct groups, based on the location of the target state within these six sub-triangles. Within each group, we select the top vertex of the sub-triangle as the designated target state and truncate the trajectories to the point at which they initially enter the target sub-triangle.

Upon completing this partitioning process for the 200 trajectories obtained from the training tasks, we arrived at six groups, each containing a respective number of trajectories: 41, 34, 27, 31, 32, 35. Within each group, we subject all four models to testing, making appropriate modifications to either the Q-function or the reward function as discussed in Sec. 2.3. To estimate the unknown parameters for each model, we employ maximum entropy IRL, optimizing the log-likelihood as defined in (6) while applying an $\mathcal{L}_1$ penalty.

To determine the coefficient for the $\mathcal{L}_1$ penalty in each model, we consider $\lambda \in \{0, 0.2, \ldots, 1\}$ and perform 5-fold cross-validation. This allows us to select the coefficient that results in the highest mean log-likelihood across the five validation sets.

Similar to Sec. 4.2, we investigate two settings for the predefined features: one that allows non-zero rewards in all states and another that restricts rewards to a subset of 8 states. Table 2 presents the AIC and BIC values (normalized by the number of observations) for different models within each group when non-zero rewards are permitted for all 81 states. It's worth noting that, while Model 2, where $Q'(s, a) = Q(s, a) + k\hat{H}(s, a)$, emerges as the best fit according to AIC for the majority of the groups, Model 4, where $Q'(s, a) = k\hat{H}(s, a)$, is selected as the best fit under the BIC criterion. This preference for Model 4 under BIC is attributed to the significant difference in the number of learned parameters between the two models, with BIC favoring the model with fewer learned parameters. Indeed, when considering non-zero rewards for all 81 states, it leads to a preference for the model with just a single learned parameter.

Table 3 presents the AIC and BIC values (normalized by the number of observations) for different models within each group when non-zero rewards are allowed for only a subset of 8 states. This setting represents a more realistic scenario with sparse rewards. Notably, in this context, Model 2 consistently emerges as the best fit according to both the AIC and BIC criteria. This suggests that humans tend to interpret evaluative feedback as a strong indicator of the long-term effectiveness of their strategic actions.

**Remark 3** *Even though Model 2 stands out as the preferred model according to both AIC and BIC criteria, there is a small evidence of support for Model 4 as well (in case of non-sparse*

**Table 2. AIC, and BIC values (normalized by the number of observations) for different models allowing non-zero rewards for all the 81 states.**

|  |  | Model 1 | | Model 2 | | Model 3 | | Model 4 | |
|---|---|---|---|---|---|---|---|---|---|
| **Num. of parameters** | | 81 | | 82 | | 82 | | 1 | |
| **Group** | **Num. of obs.** | **AIC** | **BIC** | **AIC** | **BIC** | **AIC** | **BIC** | **AIC** | **BIC** |
| **1** | 41 | 23.65 | 27.03 | 23.02 | **26.45** | 23.70 | 27.12 | **22.89** | **22.93** |
| **2** | 34 | 27.76 | 31.40 | **22.70** | **26.38** | 27.82 | 31.50 | 24.66 | **24.70** |
| **3** | 27 | 30.84 | 34.73 | **26.94** | **30.87** | 30.52 | 34.46 | 28.49 | **28.54** |
| **4** | 31 | 21.65 | 25.40 | **20.68** | **24.47** | 21.72 | 25.51 | 22.16 | **22.20** |
| **5** | 32 | 27.69 | 31.40 | **23.49** | **27.25** | 27.76 | 31.51 | 24.30 | **24.34** |
| **6** | 35 | 30.42 | 34.02 | **26.56** | **30.21** | 30.47 | 34.12 | 27.620 | **27.66** |

**Table 3. AIC, and BIC values (normalized by the number of observations) for different models allowing non-zero rewards for a sub-set of 8 states.**

| | | Model 1 | | Model 2 | | Model 3 | | Model 4 | |
|---|---|---|---|---|---|---|---|---|---|
| Num. of parameters | | 8 | | 9 | | 9 | | 1 | |
| Group | Num. of obs. | AIC | BIC | AIC | BIC | AIC | BIC | AIC | BIC |
| 1 | 41 | 31.84 | 32.18 | **22.54** | **22.92** | 31.89 | 32.27 | 22.92 | 22.96 |
| 2 | 34 | 41.72 | 42.08 | **24.03** | **24.43** | 41.77 | 42.18 | 24.76 | 24.80 |
| 3 | 27 | 44.04 | 44.43 | **27.74** | **28.17** | 44.12 | 44.55 | 28.43 | 28.48 |
| 4 | 31 | 30.49 | 30.86 | **21.26** | **21.68** | 30.56 | 30.97 | 22.20 | 22.25 |
| 5 | 32 | 42.78 | 43.15 | **23.62** | **24.03** | 42.84 | 43.26 | 24.41 | 24.45 |
| 6 | 35 | 43.53 | 43.89 | **27.25** | 27.65 | 43.59 | 43.99 | 27.57 | **27.62** |

*rewards). This suggests that there might be instances where some individuals do not primarily learn through interaction but instead focus on maximizing their evaluative feedback directly. Such individuals could potentially encounter challenges in transfer tasks where evaluative feedback is not available.*

### 4.4 Broader implications of results

Human learning and the acquisition of problem-solving skills in sequential decision-making tasks have broad implications. They can assist in cognitive rehabilitation post-injuries or strokes, enhance mathematical reasoning and STEM skill development in children, and improve performance in sports. However, mastering these skills is often challenging due to the cognitive demands of continuous decision-making. Our work introduces a systematic approach to designing advanced AI-driven tutoring systems to foster human learning in sequential decision-making tasks. As shown in Section 4.1, fostering human learning with AI-generated feedback not only promotes skill development but also facilitates the transfer of learned skills to more complex tasks. Additionally, as evidenced in Section 4.2, learning through evaluative feedback creates a more structured and organized learning experience compared to learning without feedback. Hence, these AI-based tutoring systems can improve the problem-solving skills and cognitive capabilities of the individuals while improving their learning experience.

Our findings in Section 4.3 suggest that humans perceive feedback as an indicator of the long-term effectiveness of their strategic actions. This insight can be utilized to influence human decision-making through the appropriate design of IoT devices. Specifically, by crafting feedback strategies geared towards fostering long-term behavioral enhancements, we can effectively influence individuals' long-term actions and decision-making processes.

### 5 Conclusions

In this work, we study the influence of AI-generated evaluative feedback on human decision-making, with a specific focus on sequential decision-making tasks exemplified by the Tower of Hanoi. Our study demonstrates that individuals who receive training with evaluative feedback not only experience significant improvements in their decision-making abilities but also excel in transferring these enhanced skills to similar tasks. Through an analysis utilizing the maximum entropy inverse reinforcement learning framework, we show that human learning exhibits a more structured and organized implicit reward pattern when evaluative feedback is provided during the training process. This highlights the critical role played by AI-generated feedback in improving the cognitive and strategic abilities of individuals.

Furthermore, our investigation explores various models to better comprehend how humans integrate feedback into their decision-making processes. Our findings provide substantial evidence suggesting that individuals tend to interpret evaluative feedback as a valuable indicator of the long-term effectiveness of their strategic actions. This valuable insight can be leveraged to design intelligent IoT devices, capable of enriching human learning experiences and shaping human decision-making.

## Author Contributions

**Conceptualization:** Piyush Gupta, Subir Biswas, Vaibhav Srivastava.

**Data curation:** Piyush Gupta.

**Formal analysis:** Piyush Gupta.

**Funding acquisition:** Subir Biswas, Vaibhav Srivastava.

**Investigation:** Piyush Gupta, Subir Biswas, Vaibhav Srivastava.

**Methodology:** Piyush Gupta, Subir Biswas, Vaibhav Srivastava.

**Project administration:** Piyush Gupta, Vaibhav Srivastava.

**Resources:** Piyush Gupta.

**Software:** Piyush Gupta.

**Supervision:** Subir Biswas, Vaibhav Srivastava.

**Validation:** Piyush Gupta.

**Visualization:** Piyush Gupta.

**Writing – original draft:** Piyush Gupta.

**Writing – review & editing:** Piyush Gupta, Subir Biswas, Vaibhav Srivastava.

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
