## [Decision Letter · Decision Letter 0]

15 Mar 2024

PONE-D-23-36714Fostering Human Learning in Sequential Decision-Making: Understanding the Role of Evaluative FeedbackPLOS ONE

Dear Dr. Gupta,

Thank you for submitting your manuscript to PLOS ONE. After careful consideration, we feel that it has merit but does not fully meet PLOS ONE’s publication criteria as it currently stands. Therefore, we invite you to submit a revised version of the manuscript that addresses the points raised during the review process.

Editorial Comments:

The manuscript presents an intriguing integration of methodologies traditionally not combined within a single study. It has been observed that reviewer opinions diverge considerably; however, a closer examination reveals a consensus regarding the fundamental critiques. Specifically, the manuscript draws upon methodologies from distinct academic disciplines, a factor which, when evaluated within the confines of these individual fields, might typically warrant a major revision or outright rejection. This interdisciplinary approach, while innovative, poses significant challenges for evaluation, given that most reviewers possess expertise in a singular discipline.

Despite these challenges, the interdisciplinary methodology employed by the authors holds substantial value and represents a meaningful contribution to the field. To enhance the manuscript's impact and accessibility, it is imperative for the authors to revise the manuscript in a manner that renders it comprehensible to a broader audience. Additionally, there is a substantial need for the improvement of the methods employed in the study.

In light of these considerations, the editorial decision necessitates a major revision. The authors are encouraged to undertake a comprehensive revision that addresses the aforementioned concerns, with a particular focus on making the manuscript more accessible to readers from various disciplinary backgrounds and enhancing the rigor and clarity of the methodologies employed.

We look forward to receiving your revised manuscript.

Kind regards,

Rei Akaishi

Academic Editor

PLOS ONE

 [This work has been supported in part by the NSF awards IIS-1734272 and ECCS-2024649. The analysis portion of this work is supported in part by the ONR award N00014-22-1-2813].  

Please respond by return e-mail so that we can amend your financial disclosure and competing interests on your behalf.

[This work has been supported in part by the NSF awards IIS-1734272 and ECCS-2024649. The analysis portion of this work is supported in part by the ONR award N00014-22-1-2813. ]

 [This work has been supported in part by the NSF awards IIS-1734272 and ECCS-2024649. The analysis portion of this work is supported in part by the ONR award N00014-22-1-2813.]

6. In the online submission form, you indicated that your data will be submitted to a repository upon acceptance.  We strongly recommend all authors deposit their data before acceptance, as the process can be lengthy and hold up publication timelines. Please note that, though access restrictions are acceptable now, your entire minimal  dataset will need to be made freely accessible if your manuscript is accepted for publication. This policy applies to all data except where public deposition would breach compliance with the protocol approved by your research ethics board. If you are unable to adhere to our open data policy, please kindly revise your statement to explain your reasoning and we will seek the editor's input on an exemption. 

Additional Editor Comments:

The manuscript presents an intriguing integration of methodologies traditionally not combined within a single study. It has been observed that reviewer opinions diverge considerably; however, a closer examination reveals a consensus regarding the fundamental critiques. Specifically, the manuscript draws upon methodologies from distinct academic disciplines, a factor which, when evaluated within the confines of these individual fields, might typically warrant a major revision or outright rejection. This interdisciplinary approach, while innovative, poses significant challenges for evaluation, given that most reviewers possess expertise in a singular discipline.

Despite these challenges, the interdisciplinary methodology employed by the authors holds substantial value and represents a meaningful contribution to the field. To enhance the manuscript's impact and accessibility, it is imperative for the authors to revise the manuscript in a manner that renders it comprehensible to a broader audience. Additionally, there is a substantial need for the improvement of the methods employed in the study.

In light of these considerations, the editorial decision necessitates a major revision. The authors are encouraged to undertake a comprehensive revision that addresses the aforementioned concerns, with a particular focus on making the manuscript more accessible to readers from various disciplinary backgrounds and enhancing the rigor and clarity of the methodologies employed.

Reviewers' comments:

Reviewer's Responses to Questions

**Comments to the Author**

1. Is the manuscript technically sound, and do the data support the conclusions?

Reviewer #1: Yes

Reviewer #2: Partly

2. Has the statistical analysis been performed appropriately and rigorously? 

Reviewer #1: Yes

Reviewer #2: I Don't Know

3. Have the authors made all data underlying the findings in their manuscript fully available?

Reviewer #1: Yes

Reviewer #2: No

4. Is the manuscript presented in an intelligible fashion and written in standard English?

Reviewer #1: Yes

Reviewer #2: Yes

5. Review Comments to the Author

Reviewer #1: This paper connects RL to education. I've personally been looking at connecting LLMs to education and so I enjoyed seeing this paper make these connections. From my conversations, this paper addresses a "hot topic" of mathematical thinking in education.

The statistics and experiments are clearly discussed. Personally I like using Bonferoni like conditions for the model selection (say RIC, or unbiased estimates of risk) but the methods used are fine for making the point for the paper.

Reviewer #2: Thanks for inviting me to comment on the manuscript “Fostering Human Learning in Sequential Decision-Making: Understanding the Role of Evaluative Feedback“. I am an expert in human decision making in the context of agent/individual based simulation ecological and economic models. I am not familiar with the “Tower of Hanoi” game and its substantial literature. I am also not familiar “the application of maximum entropy IRL for learning human rewards”. Therefore, I will not comment on the “maximum entropy IRL” part. I think authors should refer to “maximum entropy IRL” already in the abstract.

Summary of the parts that I address in my review: The authors study the “influence of evaluative feedback on human decision-making in sequential tasks.”. Therefore, participants solved the “Tower of Hanoi” puzzle in “five separate experiments, each featuring a different type of feedback.” “Each participant first solved a 4-disk ToH task ten times (training task) and then a 5-disk ToH task five times (transfer task) to evaluate their skill transfer to a more challenging task.”

I think the experiment could be better explained. Most of my comments below are about the experiment. Maybe a screenshot in the appendix would help to have a better intuition of the experiment. Was there a time constrain? Was time measured at all? And if so did it correlate with the success?

The feedback that participants received while playing is often called AI driven – is that appropriate for the given example? I have not really thought about it, but to me contrasting the state of the player with the optimal path is a simple calculation? I may have missed something completely here. If so the authors may want to make some less technical statements for the broad reader ship of plos one. Maybe my school of thought is also a bit restrictive identifying AI mainly with deep neural networks.

Is the sample of participants unbiased given the entry conditions: "completed a minimum of 500 prior studies and maintaining a

98% approval rate".

The authors provide no discussion. Maybe this is not to be expected in the specific research domain? Or is the discussion merged in the other parts – please comment or revise.

Specific comments:

Please explain abbreviations at their first occurrence, e.g. STEM, IoT (internet of things), RL agent…

The abstract should contain the key results of the study.

“3 Human experiments”:

Please provide numbers for m_min and m_max to allow intuition for the condition the participants have been. Did participants know the restriction in the number of allowed moves.

Is the score S visible for the participants?

The range of S is different for each experiment?

Line: 246 – how is “value of the state” defined – maybe it has been stated before, but it would be helpful to read it here again.

“3.2 Methods”: That means N = 160 and 32 participants for each experiment? Or were participants spread heterogeneously between experiments?

Lines 289 – o.k. here is part of the answer given by the authors “we collect the data of 20 participants each for the 5 set of experiments” – what happened with the remaining 60 participants?

“Results” – “percentage score” is not computed using the scores S from section “3 Human experiments” – “percentage score” may have values between 100 and ~12 assuming m_min = 15 and m_used <=m_allowed = 1.5 * m_min… I guess S_percentage is set to zero again if the participants fail to solve the puzzle? Still the visualisation by boxplots is not ideal, because they suggest that the response is continuos.

Figure 6: The authors should explain the color coding in the caption or add a label to the legend.

6. PLOS authors have the option to publish the peer review history of their article (what does this mean?). If published, this will include your full peer review and any attached files.

Reviewer #1: No

Reviewer #2: No

---

## [Author Response · Author response to Decision Letter 0]

26 Apr 2024

See attachment: Response_to_Reviewers

---

## [Decision Letter · Decision Letter 1]

3 May 2024

Fostering Human Learning in Sequential Decision-Making: Understanding the Role of Evaluative Feedback

PONE-D-23-36714R1

Dear Dr. Gupta,

We’re pleased to inform you that your manuscript has been judged scientifically suitable for publication and will be formally accepted for publication once it meets all outstanding technical requirements.

Kind regards,

Rei Akaishi

Academic Editor

PLOS ONE

Additional Editor Comments (optional):

All the issues raised by reviewers have been addressed. The careful and detailed nature of these revisions has significantly improved the manuscript, making it ready for publication.

Reviewers' comments:

Reviewer's Responses to Questions

**Comments to the Author**

1. If the authors have adequately addressed your comments raised in a previous round of review and you feel that this manuscript is now acceptable for publication, you may indicate that here to bypass the “Comments to the Author” section, enter your conflict of interest statement in the “Confidential to Editor” section, and submit your "Accept" recommendation.

Reviewer #1: All comments have been addressed

Reviewer #2: (No Response)

2. Is the manuscript technically sound, and do the data support the conclusions?

Reviewer #1: Yes

Reviewer #2: Yes

3. Has the statistical analysis been performed appropriately and rigorously? 

Reviewer #1: Yes

Reviewer #2: I Don't Know

4. Have the authors made all data underlying the findings in their manuscript fully available?

Reviewer #1: Yes

Reviewer #2: Yes

5. Is the manuscript presented in an intelligible fashion and written in standard English?

Reviewer #1: Yes

Reviewer #2: Yes

6. Review Comments to the Author

Reviewer #1: I'm happy with this version and I have no further comments. I didn't have any major complaints on the first round, so this revision mostly addresses the other reviewer and editor's comments.

Reviewer #2: Thanks for allowing me to have a look at the revision of the manuscript “Fostering human learning in sequential decision-making: Understanding the role of evaluative feedback”. Reading through the manuscript again and the comments by the authors and the editor was really insightful. All my comments have been treated appropriately. I had a very quick look through the repository and was wondering whether the authors want to state in the main text that matlab has been used for the analysis. Thanks to everyone involved in this process.

7. PLOS authors have the option to publish the peer review history of their article (what does this mean?). If published, this will include your full peer review and any attached files.

Reviewer #1: No

Reviewer #2: No

---

## [Editor Report · Acceptance letter]

7 May 2024

PONE-D-23-36714R1 

PLOS ONE

Dear Dr. Gupta, 

I'm pleased to inform you that your manuscript has been deemed suitable for publication in PLOS ONE. Congratulations! Your manuscript is now being handed over to our production team.

Kind regards, 

on behalf of

Dr. Rei Akaishi 

Academic Editor

PLOS ONE